# A Gold Standard Dataset for the Reviewer Assignment Problem

## Abstract

Many peer-review venues are either using or looking to use algorithms to assign submissions to reviewers. The crux of such automated approaches is the notion of the "*similarity score*"—a numerical estimate of the expertise of a reviewer in reviewing a paper—and many algorithms have been proposed to compute these scores. However, these algorithms have not been subjected to a principled comparison, making it difficult for stakeholders to choose the algorithm in an evidence-based manner. The key challenge in comparing existing algorithms and developing better algorithms is the lack of the publicly available gold-standard data that would be needed to perform reproducible research. We address this challenge by collecting a novel dataset of similarity scores that we release to the research community. Our dataset consists of 477 self-reported expertise scores provided by 58 researchers who evaluated their expertise in reviewing papers they have read previously.

We use this data to compare several popular algorithms currently employed in computer science conferences and come up with recommendations for stakeholders. Our three main findings are:

- All algorithms make a non-trivial amount of error. For the task of ordering two papers in terms of their relevance for a reviewer, the error rates range from 12%-30% in easy cases to 36%-43% in hard cases, thereby highlighting the vital need for more research on the similarity-computation problem.
- Most existing algorithms are designed to work with titles and abstracts of papers, and in this regime the SPECTER+MFR algorithm performs best.
- To improve performance, it may be important to develop modern deep-learning based algorithms that can make use of the full texts of papers: the classical TD-IDF algorithm enhanced with full texts of papers is on par with the deep-learning based SPECTER+MFR that cannot make use of this information.

We encourage researchers to use this dataset for evaluating and developing better similarity-computation algorithms.

## 1   Introduction

Assigning papers to reviewers with appropriate expertise is the key prerequisite for high-quality reviews (Thurner & Hanel, 2011; Black et al., 1998; Bianchi & Squazzoni, 2015). Even a small fraction of incorrect reviews can negatively impact the quality of the published scientific standard (Thurner & Hanel, 2011) and hurt careers of researchers (Merton, 1968; Squazzoni & Gandelli, 2012; Thorngate & Chowdhury, 2014). Quoting Triggle & Triggle (2007):

> *"An incompetent review may lead to the rejection of the submitted paper, or of the grant application, and the ultimate failure of the career of the author."*

Conventionally, the selection of reviewers for a submission was the task of a journal editor or a program committee of a conference. However, the rapid growth in the number of submissions to many publication

venues (Shah, 2022) has made the manual selection of reviewers extremely challenging. As a result, many peer-review venues in computer science as well as other fields are either using or looking to use algorithms to assist organizers in assigning submissions to reviewers (Garg et al., 2010; Charlin & Zemel, 2013; Kobren et al., 2019; Kerzendorf et al., 2020; Stelmakh et al., 2021; OpenReview, 2022).

The key component of existing automated approaches for assigning reviewers to submissions is the "*similarity score*". For any reviewer-submission pair, the similarity score is a number that captures the expertise of the reviewer in reviewing the submission. The assignment process involves first computing the similarity score for each submission-reviewer pair, and then using these scores to assign the reviewers to submissions (Shah, 2022, Section 3). The similarity scores are typically computed by matching the text of the submission with the profile (e.g., past papers) of the reviewer, which is the primary focus of this paper. Additionally, these scores may also be augmented by manually selected subject areas or reviewer-provided manual preferences ("bids").

Several algorithms for computing similarity scores have been already proposed and used in practice (we review these algorithms in Sections 2 and 5). However, there are two key challenges in designing and using such algorithms in a principled manner. First, there is no publicly available gold standard data that can be used for algorithm development. Second, despite the existence of many similarity-computation algorithms, the absence of the gold standard data prevents principled comparison of these algorithms. As a result, three flagship machine learning conferences—ICML, NeurIPS, and ACL—rely on three different similarity-computation algorithms, and the differences in performance of these algorithms are not well understood.

In this work, *we address the aforementioned challenges and collect a dataset of reviewers' expertise that can facilitate the progress in the reviewer assignment problem.* Specifically, we conduct a survey of computer science researchers whose experience level ranges from graduate students to senior professors. In the survey, we ask participants to report their expertise in reviewing computer science papers they read over the last year.

**Contributions**   Overall, our contributions are threefold:

- First, we collect and release a high-quality dataset of reviewers' expertise that can be used for training and/or evaluation of similarity-computation algorithms. The dataset can be found attached as supplementary material, and will be publicly posted on GitHub along with the camera ready version of the paper.

- Second, we use the collected dataset to compare existing similarity-computation algorithms and inform organizers in making a principled choice for their venue. Specifically, we observe that when all algorithms operate with titles and abstracts of papers, the most advanced SPECTER+MFR algorithm performs best. However, when the much simpler TPMS is additionally provided with full texts of papers, it achieves the same level of performance as SPECTER+MFR.

- Third and finally, we conduct an exploratory analysis that highlights areas of improvement for existing algorithms and our insights can be used to develop better algorithms to improve peer review. For example, we believe that an important direction is to develop deep-learning algorithms that employ full texts of papers to improve performance.

Overall, we observe that all current algorithms exhibit non-trivial amounts of error. When tasked to compare a pair of papers in terms of the expertise of a given reviewer, all algorithms make 12%-30% mistakes even when two papers are selected to have a large difference in reviewer's expertise. On a more challenging task of comparing two high-expertise papers, the algorithms err with probability 36%-43%. This observation underscores the vital need to design better algorithms for matching reviewers to papers.

Let us now make three important remarks. First, while our dataset comprises researchers and papers from computer science and closely-related fields, other communities may also use it to evaluate existing or develop new similarity-computation algorithms. To evaluate an algorithm from another domain on our data, researchers can fine-tune their algorithm on profiles of computer science scientists crawled from Semantic Scholar and then evaluate it on our dataset.

Second, as we discuss later in more detail, the dataset is not devoid of biases, with bias in terms of geographical skew of participants being most prominent.

This leads to the third remark: in this work we release an *initial* version of the dataset consisting of 477 data points contributed by 58 researchers. Our dataset is not set in stone and we encourage researchers to participate in our survey and contribute their data to the dataset. By collecting more samples, we enable more fine-grained comparisons and also improve the diversity of the dataset in terms of both population of participants and subject areas of papers. The survey is available at:

*(link redacted for double blind reviewing; survey replicated in Appendix A)*

and we will be updating the released version regularly.

## 2 Related literature

In this section, we discuss relevant past studies. We begin with an overview of works that report comparisons of different similarity-computation algorithms. We then provide a brief discussion of the commonly used procedure for computing similarity scores, which are ultimately utilized to assign reviewers to submissions. Finally, we conclude with a list of works that design algorithms to automate other aspects of reviewer assignment.

**Evaluation of similarity-computation algorithms**  The OpenReview platform (OpenReview, 2022) uses an approach of predicting authorship as a proxy to measuring the quality of the similarity scores computed by any algorithm. Specifically, they consider papers authored by a number of researchers, remove one of these papers from the corpus, and predict expertise of each researcher in reviewing the selected paper. The performance of an algorithm then is measured as a fraction of times the author of the selected paper is predicted to be among the top reviewers for this paper. This authorship proxy, however, may not be representative of the actual task of similarity computation as algorithms that accurately predict the authorship relationship (and hence do well according to this approach) are not guaranteed to accurately estimate expertise in reviewing submissions authored by other researchers.

Mimno & McCallum (2007) collected a dataset with external expertise judgments. Specifically, they used 148 papers accepted to the NeurIPS 2006 conference and 364 reviewers from the NeurIPS 2005 conference and ask human annotators – independent established researchers – to evaluate expertise for a selected subset of 650 (paper, reviewer) pairs. Similarly, Zhao et al. (2022) collected the set of papers and reviewers in the ICIP 2016 conference. They then asked independent researchers to provide their estimates of the match between various reviewer-paper pairs. While this approach results in a publicly available dataset, we note that external expertise judgments may also be noisy as judges may have incomplete information about the expertise of reviewers.

Dumais & Nielsen (1992), Rodriguez & Bollen (2008) and Anjum et al. (2019) obtain more accurate expertise judgments by relying on self reports of expertise evaluations from reviewers. In more detail, Dumais & Nielsen (1992) and Rodriguez & Bollen (2008) rely on *ex-ante* bids—preferences of reviewers in reviewing submissions made in advance of reviewing. In contrast, Anjum et al. (2019) rely on *ex-post* evaluations of expertise made by reviewers after reviewing the submissions. These works construct datasets that can be used to evaluate algorithms: Dumais & Nielsen (1992) employ 117 papers and 15 reviewers, Rodriguez & Bollen (2008) employ 102 papers and 69 reviewers and Anjum et al. (2019) employ 20 papers and 33 reviewers. However, Rodriguez & Bollen (2008) and Anjum et al. (2019) use sensitive data that cannot be released without compromising the privacy of reviewers. Furthermore, *ex-ante* evaluations of Dumais & Nielsen (1992) and Rodriguez & Bollen (2008) may not have a high accuracy as (i) bids may contaminate expertise judgments with *willingness* to review submissions, and (ii) bids are based on a very brief acquaintance with papers. On the other hand, *ex-post* data by Anjum et al. (2019) is collected for papers assigned to reviewers using a specific similarity-computation algorithm. Thus, while collected evaluations have high precision, they may also have low recall if the employed similarity-computation algorithm erroneously assigned low expertise scores to some (paper, reviewer) pairs as evaluations of expertise for such papers were not observed.

In this work, we collect a novel dataset of reviewer expertise that (i) can be released publicly, and (ii) contains accurate self-evaluations of expertise that are based on a deep understanding of the paper and are not biased towards any existing similarity-computation algorithm.

**Similarity scores in conferences**  In modern conferences, similarity scores are typically computed by combining two types of input:

- *Initial automated estimates.* First, a similarity-computation algorithm is used to compute initial estimates. Many algorithms have been proposed for this task (Mimno & McCallum, 2007; Rodriguez & Bollen, 2008; Charlin & Zemel, 2013; Liu et al., 2014; Tran et al., 2017; Anjum et al., 2019; Kerzendorf et al., 2020; OpenReview, 2022) and we provide more details on several algorithms used in flagship computer science conferences in Section 5.

- *Human corrections.* Second, automated estimates are corrected by reviewers who can read abstracts of submissions and report bids—preferences in reviewing the submissions. A number of works focus on the bidding stage and (i) explore the optimal strategy to assist reviewers in navigating the pool of thousands of submissions (Fiez et al., 2020; Meir et al., 2020) or (ii) protect the system from strategic bids made by colluding reviewers willing to get assigned to each other's paper (Jecmen et al., 2020; Wu et al., 2021; Boehmer et al., 2021; Jecmen et al., 2022).

Combining these two types of input in a principled manner is a non-trivial task. As a result, different conferences use different strategies (Shah et al., 2018; Leyton-Brown et al., 2022) and there is a lack of empirical or theoretical evidence that would guide venue organizers in their decisions.

**Automation of the assignment stage**  At a high level, automated assignment stage consists of two steps: first, similarity scores are computed; second, reviewers are allocated to submissions such that some notion of assignment quality (formulated in terms of the similarity scores) is maximized. In this work, we focus on the first step of the process. However, for completeness, we now mention several works that design algorithms for the second step.

A popular notion of assignment quality is the cumulative similarity score, that is, the sum of the similarity scores across all assigned reviewers and papers. An algorithm pursuing such an objective is implemented in the widely employed TPMS assignment algorithm (Charlin & Zemel, 2013) and similar ideas are explored in many papers (Goldsmith & Sloan, 2007; Tang et al., 2010; Long et al., 2013). While the cumulative objective is a natural choice, it has been noted that it may discriminate against certain submissions by allocating all irrelevant reviewers to a subset of submissions, even when a more balanced assignment exists (Garg et al., 2010). Thus, a number of works has explored the idea of assignment fairness, aiming at producing more balanced assignments (Kobren et al., 2019; Stelmakh et al., 2021). Finally, other works explore the ideas of envy-freeness (Tan et al., 2021; Payan, 2022), resistance to lone-wolf strategic behavior (Xu et al., 2019; Dhull et al., 2022), and encouraging various types of diversity (Li et al., 2015; Leyton-Brown et al., 2022).

## 3   Data collection pipeline

In this section, we describe the process of data collection. This work was performed under the approval of an Institutional Review Board (IRB).

**Gold standard data**  In this study, we aim to collect a dataset of self-evaluations of reviewers' expertise that satisfies two desiderata:

(D1) The dataset should comprise evaluations of expertise for papers participants read to a reasonable extent.

(D2) The dataset should be released publicly without disclosing any sensitive information.

Let us now discuss our approach to recruiting participants and obtaining accurate estimates of their expertise in reviewing papers included in the dataset.

**Participant recruiting**   We recruited participants using a combination of several channels that are typically employed to recruit reviewers for computer science conferences:

- *Mailing lists.* First, we sent recruiting emails to relevant mailing lists of several universities and research departments of companies.

- *Social media.* Second, via a call for participation on X (formerly Twitter).

- *Personal communication.* Third, we sent personal invites to researchers from our professional network.

We had a screening criterion requiring that prospective participants have at least one paper published in the broad area of computer science. Overall, for the version of the dataset we release in this paper, we managed to recruit 58 participants, all of whom passed the screening.

**Expertise evaluations**   The key idea of our approach to expertise evaluation is to ask participants to *evaluate their expertise in reviewing papers they read in the past.* After reading a paper, a researcher is in the best possible position to evaluate whether they have the right background—both in terms of the techniques used in the paper and in terms of the broader research area of the paper—to judge the quality of the paper. With this motivation, we asked participants to:

> *Recall 5-10 papers in their broad research area that they read to a reasonable extent in the last year and tell us their expertise in reviewing these papers.*

In more detail, the choice of papers was constrained by two minor conditions:

- The papers reported by a participant should not be authored by them.

- The papers reported by a participant should be freely available online.

In addition to these constraints, we gave several recommendations to the participants in order to make the dataset more diverse and useful for the research purposes:

- First, we asked participants to choose papers that cover the whole spectrum of their expertise with some papers being well-separated (e.g., very high expertise and very low expertise) and some papers being nearly-tied (e.g., two high-expertise papers).

- Second, we recommended participants to avoid ties in their evaluations. To help participants comply with this recommendation, we implemented evaluation on a scale of 1 ("I am not qualified to review this paper") to 5 ("I have background necessary to evaluate all the aspects of the paper") with a 0.25 step size, enabling participants to report papers with small differences in expertise.

- Third, we asked participants to come up with papers that they think may be tricky for existing similarity-computation algorithms. For this, we relied on the commonsense understanding and did not instruct participants on the inner-workings of these algorithms. We only provided an example: *a naive computation of similarity may think that a paper on "Theory of Information Dissemination in Social Networks" has high similarity with an Information Theory researcher, but in reality, this researcher may not have expertise in reviewing this paper.*

Overall, the time needed for participants to contribute to the dataset was estimated to be 5–10 minutes. The full instructions of the survey are available in Appendix A.

**Data release**   Following the procedure outlined above, we collected responses from 58 researchers. These responses constitute an initial version of the dataset that we release in this work. Each entry in the dataset corresponds to a participant and comprises evaluations of their expertise in reviewing papers of their choice. For each paper and each participant, we provide representations that are sufficient to start working on our dataset:

TOTAL NUMBER OF PARTICIPANTS: 58

| CHARACTERISTIC | QUANTITY | VALUE |
|---|---|---|
| GENDER | % MALE | 78 |
| COUNTRY | % USA | 74 |
| POSITION | % PhD STUDENT | 45 |
| | % FACULTY | 28 |
| | % POST-PhD (NON-FACULTY) | 12 |
| EXPERIENCE | MIN # PUBLICATIONS | 2 |
| | MAX # PUBLICATIONS | 492 |
| | MEAN # PUBLICATIONS | 54 |
| | MEDIAN # PUBLICATIONS | 20 |

Table 1: Demography of participants. For the first four characteristics, quantities represent percentages of the one or more most popular classes in the dataset. Note that classification is done manually based on publicly available information and may not be free of error. For the last characteristic, quantities are computed based on Semantic Scholar profiles.

- *Participant.* Each participant is represented by their Semantic Scholar ID and complete bibliography crawled from Semantic Scholar on May 1, 2022.

- *Paper.* Each paper, including papers from participants' bibliographies, is represented by its Semantic Scholar ID, title, abstract, list of authors, publication year, and arXiv identifier. Additionally, papers from participants' responses are supplied with links to freely available PDFs (whenever available).

## 4    Data exploration

In this section we explore the collected data and present various characteristics of the dataset. The subsequent sections then detail the results of using this data to benchmark various popular similarity-computation algorithms.

### 4.1    Participants

We begin with a discussion of the pool of the survey participants: Table 1 displays its key characteristics. First, we note that all participants work in the broad area of computer science and have rich publication profiles (at least two papers published, with the mean of 54 papers and the median of 20 papers). In many subareas of computer science, including machine learning and artificial intelligence, having two papers is usually sufficient to join the reviewer pool of flagship conferences. Given that approximately 85% of participants either have PhD or are in the process of getting the degree, we conclude that most of the researchers who contributed to our dataset are eligible to review for machine learning and artificial intelligence conferences.

Second, we caveat that most of the participants are male researchers affiliated with US-based organizations, with about 40% of all participants being affiliated with a certain university(redacted for double blind reviewing). Hence, the population of participants is not necessarily representative of the general population of the machine learning and computer science communities. We encourage researchers to be aware of this limitation when using our dataset. We note that the data collection process does not conclude with the publication of the present paper and we will be updating the dataset as new responses come. We also encourage readers to contribute 5–10 minutes of their time to fill out the survey *(link redacted for double blind reviewing; survey replicated in Appendix A)* and make the dataset more representative.

TOTAL NUMBER OF PAPERS: 463

| CHARACTERISTIC | QUANTITY | VALUE |
|---|---|---|
| OPEN ACCESS | # ON SEMANTIC SCHOLAR | 462 |
| | # ON ARXIV | 411 |
| | # PDF AVAILABLE | 457 |
| RESEARCH AREAS | # COMPUTER SCIENCE | 459 |
| PUBLICATION YEAR | % BEFORE 2020 | 25 |
| | % 2020 OR LATER | 75 |

Table 2: Characteristics of the 463 papers in the released dataset. Most of the papers are freely available online and belong to the broad area of computer science.

## 4.2 Papers

We now describe the set of papers that constitute our dataset. Overall, participants evaluated their expertise in reviewing 463 unique papers. Out of these 463 papers, 12 papers appeared in reports of two participants, 1 paper was mentioned by three participants, and the remaining papers were mentioned by one participant each.

Table 2 presents several characteristics of the pool of papers in our dataset. First, we note that all but one of the papers are listed on Semantic Scholar, enabling similarity-computation algorithms developed on the dataset to query additional information about the papers from the Semantic Scholar database. For the paper that does not have a Semantic Scholar profile, we construct such a profile manually and keep this paper in the dataset. Additionally, most of the papers (99%) have their PDFs freely available online, thereby allowing algorithms to use full texts of papers to compute similarities.

Semantic Scholar has a built-in tool to identify broad research areas of papers (Wade, 2022). According to this tool, 99% of the papers included to our dataset belong to the broad area of computer science—the target area for our data-collection procedure. The remaining four papers belong to the neighboring fields of mathematics, philosophy, and the computational branch of biology. Finally, approximately 75% of all papers in our dataset are published in or after 2020, ensuring that our dataset contains recent papers that similarity-computation algorithms encounter in practice.

## 4.3 Evaluations of expertise

Finally, we proceed to the key aspect of our dataset—evaluations of expertise in reviewing the papers reported by participants. All but one participant reported expertise in reviewing at least 5 papers with the mean number of papers per participant being 8.2 and the total number of expertise evaluations being 477.

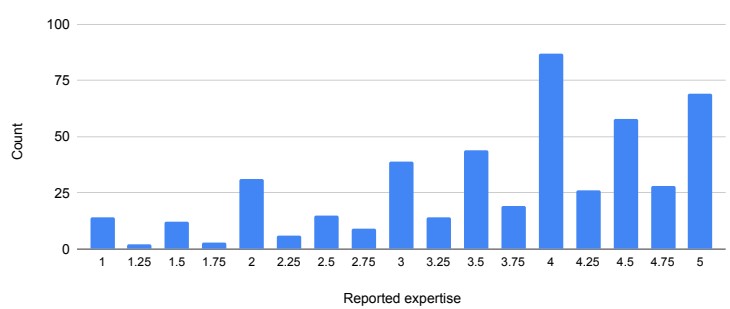

(a) Counts of expertise values reported by participants.

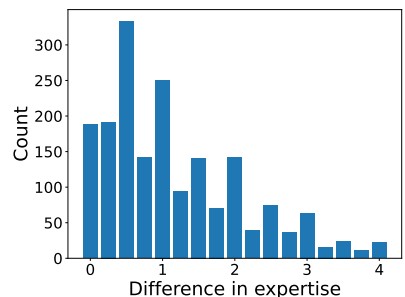

(b) Counts of differences in expertise evaluations.

Figure 1: Distribution of expertise scores reported by participants.

Figure 1 provides visualization of expertise evaluations made by participants. First, Figure 1a displays the counts of expertise values. About 56% of the reported papers have an expertise score greater than equal to 4, thereby yielding a good spread.

Second, Figure 1b shows the distributions of pairwise differences in expertise evaluations made by the same reviewer. To construct this figure, for each participant we considered all pairs of papers in their report. Next, we pooled the absolute values of the pairwise differences in expertise across participants. We then plotted the histogram of these differences in the figure. Observe that the distribution in Figure 1b has a heavy tail, suggesting that our dataset is suitable for evaluating the accuracy of similarity-computation methods both at a coarse level (large differences between the values of expertise) and at a fine level (small differences between the values of expertise).

## 5 Experimental setup

We now describe the setup of experiments on our dataset.

### 5.1 Metric

We begin with defining a main metric that we use in this work to evaluate performance of the algorithms. For this, we rely on the Kendall's Tau distance that is closely related to the widely used Kendall's Tau (Kendall, 1938) rank correlation coefficient. We introduce the metric and the algorithms in this section, followed by the results in Section 6. Subsequently in Section 7, we provide additional evaluations separating out hard and easy instances.

**Intuition**   Before we introduce the metric in full details, let us provide some intuition behind it. Consider a pair of papers and a participant who evaluated their expertise in reviewing these papers. We say that a similarity-computation algorithm makes an error if it fails to correctly predict the paper for which the participant reported the higher value of expertise.

Of course, some pairs of papers are harder to resolve than others (e.g., it is harder to resolve papers with expertise scores 4.0 and 4.25 than 4.0 and 1.0). To capture this intuition, whenever an algorithm makes an error, we penalize it by *the absolute difference in the expertise values reported by the participant.* Overall, the loss of the algorithm is the sum of losses across all pairs of papers evaluated by participants normalized to take values between 0 and 1 (the lower the better).

**Formal definition**   The intuition behind the metric that we have just introduced should be sufficient to interpret the results of our study so the reader may skip the rest of this section and move directly to Section 5.2. However, for the sake of rigor, we now introduce our metric more formally.

Consider any algorithm that produces real-valued predictions of reviewers' expertise in reviewing a given set of papers. We call these predictions the "similarity scores" given by the algorithm. We assume that a higher value of a similarity score means that the algorithm predicts a better expertise. The metric we define below is agnostic to the range or exact values of predicted similarity scores; it only relies on the relative values across different papers.

Now consider any participant $r$ in our study. We let $m_r$ denote the number of papers for which participant $r$ reports their expertise. We let $\mathcal{P}_r = \{p_r^{(1)}, p_r^{(2)}, \ldots, p_r^{(m_r)}\}$ denote this set of $m_r$ papers. For every $i \in \{1, \ldots, m_r\}$, we let $\varepsilon_r^{(i)} \in \{1, 1.25, 1.5, \ldots, 5\}$ denote the expertise self-reported by participant $r$ for paper $p_r^{(i)}$. Next, for every $i \in \{1, \ldots, m_r\}$, we let $s_r^{(i)}$ denote the real-valued similarity score given by the algorithm to the pair (reviewer $r$, paper $p_r^{(i)}$).

Having set up this notation, we now define the 'unnormalized' loss of the algorithm with respect to participant $r$ as:

$$L_r = \sum_{\substack{i,j=1 \\ i<j}}^{m_r} \left( \underbrace{\mathbb{I}\left\{ (s_r^{(i)} - s_r^{(j)}) \times (\varepsilon_r^{(i)} - \varepsilon_r^{(j)}) < 0 \right\}}_{\text{error}} \times \left| \varepsilon_r^{(i)} - \varepsilon_r^{(j)} \right| + \underbrace{\mathbb{I}\left\{ (s_r^{(i)} - s_r^{(j)}) \times (\varepsilon_r^{(i)} - \varepsilon_r^{(j)}) = 0 \right\}}_{\text{tie}} \times \frac{1}{2} \left| \varepsilon_r^{(i)} - \varepsilon_r^{(j)} \right| \right).$$

In words, for each pair of papers $(p_r^{(i)}, p_r^{(j)})$ reported by participant $r$, the algorithm is not penalized when the ordering of papers induced by the similarity scores $\{s_r^{(i)}, s_r^{(j)}\}$ agrees with the ground truth expertise-based ordering $\{\varepsilon_r^{(i)}, \varepsilon_r^{(j)}\}$. When two orderings disagree (that is, the algorithm makes an error), the algorithm is penalized by the difference of expertise reported by the participant ($|\varepsilon_r^{(i)} - \varepsilon_r^{(j)}|$). Finally, when scores computed by the algorithm indicate a tie while expertise scores are different, the algorithm is penalized by half the difference in expertise ($\frac{1}{2}|\varepsilon_r^{(i)} - \varepsilon_r^{(j)}|$).

Having the unnormalized loss with respect to a participant defined, we now compute the overall loss $L \in [0, 1]$ of the algorithm. For this, we take the sum of unnormalized losses across all participants and normalize this sum by the loss achieved by the adversarial algorithm that reverses the ground-truth ordering of expertise (that is, sets $s = -\varepsilon$) and achieves the worst possible performance on the task. More formally,

$$L = \frac{\sum\limits_{r} L_r}{\sum\limits_{r} \sum\limits_{\substack{i,j=1 \\ i<j}}^{m_r} \left| \varepsilon_r^{(i)} - \varepsilon_r^{(j)} \right|}.$$

Overall, our loss $L$ takes values from 0 to 1 with lower values indicating better performance.

## 5.2 Algorithms

In this work, we evaluate several algorithms that we now discuss. All of these algorithms operate with (i) the list of submissions for which similarity scores need to be computed and (ii) reviewers' profiles comprising past publications of reviewers. Conceptually, all algorithms predict reviewers' expertise by evaluating textual overlap between each submission and papers in each reviewer's publication profile. Let us now provide more detail on how this idea is implemented in each of the algorithms under consideration.

**Trivial baseline** First, we consider a trivial baseline that ignores the content of submissions and reviewers' profiles when computing the assignment scores: for each (participant, paper) pair, the TRIVIAL algorithm predicts the score $s$ to be 1.

**Toronto Paper Matching System (TPMS)** The TPMS algorithm (Charlin & Zemel, 2013), which is based on TF-IDF similarities, is widely used by flagship conferences such as ICML and AAAI. While an exact implementation is not publicly available, in our experiments we use an open-source version by Xu et al. (2019) which implements the basic TF-IDF logic of TPMS. We also note that the TF-IDF method was also independently proposed earlier by Price et al. (2010), and subsequently also rediscovered by Kerzendorf et al. (2020).

As a technical remark, in our implementation we use reviewers' profiles and all reported papers to compute the IDF part of the TF-IDF model. In principle, one may be able to get a better performance by using a larger set of papers from the respective field (e.g., all submissions to the last edition of the conference) to compute IDF.

**OpenReview algorithms** OpenReview is a conference-management system used by machine learning conferences such as NeurIPS and ICLR. It offers a family of deep-learning algorithms for measuring expertise of reviewers in reviewing submissions. In this work, we evaluate the following algorithms which are used to compute affinity scores between submissions and reviewers:

- ELMo. This algorithm relies on general-purpose **E**mbeddings from **L**anguage **Mo**dels (Peters et al., 2018) to compute textual similarities between submissions and reviewers' past papers.

- SPECTER. This algorithm employs more specialized document-level embeddings of scientific documents (Cohan et al., 2020). Specifically, SPECTER explores the citation graph to construct embeddings that are useful for a variety downstream tasks, including the similarity computation we focus on in this work.

- SPECTER+MFR. Finally, SPECTER+MFR further enhances SPECTER (Chang & McCallum, 2021). Instead of constructing a single embedding of each paper, it constructs multiple embeddings that correspond to different facets of the paper. These embeddings are then used to compute the similarity scores.

We use implementations of these methods that are available on the OpenReview GitHub page[1] and execute them with default parameters.

**ACL paper matching**  The Association for Computational Linguistics (ACL) is a community of researchers working on computational problems involving natural language. The association runs multiple publication venues, including the flagship ACL conference, and has its own method to compute expertise between papers and reviewers.[2] This algorithm is trained on 45,309 abstracts of papers found in the ACL anthology (`https://aclanthology.org`) downloaded in November 2019. The key idea of the model training is to split each abstract into two non-contiguous parts and treat two parts of the same abstract as a positive example and highly similar parts of different abstracts as negative examples. The algorithm uses ideas from the work of Wieting et al. (2019; 2022) to learn a simple and efficient similarity function that separates positive examples from negative examples. This function is eventually used to compute similarity scores between papers and reviewers. More details on the ACL algorithm are provided in Appendix B.

We note that the ACL algorithm is trained on the domain of computational linguistics and hence may suffer from a distribution shift when applied to the general machine learning domain. That said, in line with ELMo, SPECTER, and SPECTER+MFR, we use the ACL algorithm without additional fine-tuning.

## 6  Results

In this section, we report the results of evaluation of algorithms described in Section 5.2. First, we juxtapose all algorithms on our data (Section 6.1). Second, we use the TPMS algorithm to explore various aspects of the similarity-computation problem (Section 6.2).

Before presenting the results, we note that in this section, we conduct all evaluations focusing on papers that have PDFs freely available. To this end, we remove 6 papers from the dataset as they are not freely available online (see Table 2). Similarly, we limit reviewer profiles to papers whose semantic scholar profiles contain links to arXiv. One of the participants did not have any such papers, so we exclude them from the dataset.

### 6.1  Comparison of the algorithms

Our first set of results compares the performance of the existing similarity-computation algorithms. To run these algorithms on our data, we need to make some modeling choices faced by conference organizers in practice:

- *Paper representation.* First, in their inner-workings, similarity-computation algorithms operate with some representation of the paper content. Possible choices of representations include: (i) title of the paper, (ii) title and abstract, and (iii) full text of the paper. We choose option (ii) as this option is often used in real conferences and is supported by all algorithms we consider in this work. Thus, to predict

---

[1] `https://github.com/openreview/openreview-expertise`
[2] `https://github.com/acl-org/reviewer-paper-matching`

expertise, algorithms are provided with the title and abstract of each paper (both papers submitted to the conference and papers in reviewers' publication profiles).

- *Reviewer profiles.* The second important parameter is the choice of papers to include in reviewers' profiles. In real conferences, this choice is often left to reviewers who can manually select the papers they find representative of their expertise. In our experiments, we construct reviewer profiles automatically by using the 20 most recent papers from their Semantic Scholar profiles. If a reviewer has less than 20 papers published, we include all of them in their profile. Our choice of the reviewer profile size is governed by the observation that the mean length of the reviewer profile in the NeurIPS 2022 conference is 16.5 papers. By setting the maximum number of papers to 20, we achieve the mean profile length of 14.8, thereby operating with the amount of information close to that available to algorithms in real conferences.

**Statistical aspects**  To build reviewer profiles, we use publication years to order papers by recency, where we break ties uniformly at random. Thus, the content of reviewer profiles depends on randomness. To average this randomness out, we repeat the procedure of profile construction and similarity prediction 10 times, and report the mean loss over these iterations. That said, we note that the observed variability due to the randomness in the construction of reviewer profiles is negligible, with a standard deviation over all iterations is less than 0.005.

The pointwise performance estimates obtained by the procedure above depend on the selection of participants who contributed to our dataset. To quantify the associated level of uncertainty, we compute 95% confidence intervals as follows. For 1,000 iterations, we create a new *reviewer pool* by sampling participants with replacement and recomputing the loss of each algorithm on the bootstrapped set of reviewers. To save computation time, we do not reconstruct reviewer profiles for each of these iterations as the uncertainty associated with the construction of reviewer profiles is small. Instead, we reuse profiles constructed to obtain pointwise estimates.

Finally, we also build confidence intervals for the *difference* in the performance of the algorithms. We do so in addition to the aforementioned procedure since even when the losses of the algorithms fluctuate with the choice of the bootstrapped dataset, the relative difference in performance of a pair of algorithms may be stable. Specifically, we use the procedure above to build confidence intervals for the difference in performance between the TPMS algorithm and each of the more advanced algorithms. TPMS is chosen as a baseline for this comparison due to its simplicity.

**Results of the comparison**  Table 3 displays results of the comparison. The first pair of columns presents the loss of each algorithm on our dataset and the associated confidence intervals. The third and the forth columns investigate the relative difference in performance between the non-trivial algorithms: they display the differences in performance between TPMS and each of the more advanced algorithms (OpenReview algorithms and ACL) together with the associated confidence intervals. We make several important observations from Table 3:

- First, we note that all algorithms we consider in this work considerably outperform the Trivial baseline, confirming that content of papers is useful in evaluating expertise.

- Second, comparing three algorithms from the OpenReview toolkit, we note that Specter+MFR and Specter outperform ELMo. The former two algorithms rely on domain-specific embeddings[3] while ELMo uses general-purpose embeddings. Thus, the nature of the text similarity computation task in the academic context may be sufficiently different from that in other domains.

- Third, we note that under modeling choices (paper representation and length of reviewers' profiles) that mimic choices that have been made in real conferences, the Specter+MFR algorithm performs best.

- The fourth finding is the most surprising. Observe that TPMS algorithm is much simpler than other non-trivial algorithms: in contrast to ELMo, Specter, Specter+MFR, and ACL, it does not rely on carefully learned embeddings. However, TPMS is competitive against complex Specter

---

[3]Specter+MFR and Specter are initialized with SciBERT Beltagy et al. (2019)

| Algorithm | Loss | 95% CI for Loss | Δ with TPMS | 95% CI for Δ |
|-----------|------|-----------------|-------------|--------------|
| Trivial | 0.50 | — | — | — |
| TPMS | 0.28 | $[0.23, 0.33]$ | — | — |
| ELMo | 0.35 | $[0.31, 0.42]$ | $+0.07$ | $[0.02, 0.14]$ |
| Specter | 0.27 | $[0.21, 0.34]$ | $-0.01$ | $[-0.06, 0.04]$ |
| Specter+MFR | **0.24** | $[0.18, 0.30]$ | $-0.04$ | $[-0.09, 0.01]$ |
| ACL | 0.30 | $[0.25, 0.36]$ | $+0.02$ | $[-0.02, 0.07]$ |

Table 3: Comparison of similarity-computation algorithms on the collected data. All algorithms operate with reviewer profiles consisting of the 20 most recent papers and use titles and abstracts of papers. Lower values of loss are better. A positive (respectively, negative) value of Δ indicates that the algorithm performs worse (respectively, better) than TPMS.*

*In Table 4 we evaluate TPMS algorithm when full texts of papers are additionally provided and observe that TPMS closes the gap with Specter+MFR when using this information.

and Specter+MFR and even outperforms ELMo and ACL. Moreover, TPMS is the only algorithm that can make use of the full text of the papers. In Section 6.2, we find that when TPMS uses the full text of all papers (including reviewers' past papers), it closes the gap with Specter+MFR.

Note that the performance of advanced algorithms could be improved by fine-tuning in a dataset-specific manner, or by leveraging larger pre-trained models like T5 (Raffel et al., 2020) with added fine-tuning in the scientific domain. However, our observation suggests that the off-the-shelf performance of the best algorithm (Specter+MFR) is comparable to that of much simpler TPMS algorithm.

Finally, we note that some of the confidence intervals for the performance of the different algorithms, as well as for the relative differences, are overlapping. It is, therefore, crucial to increase the size of our dataset to enable more fine-grained comparisons between the algorithms.

## 6.2 The role of modeling choices

In the beginning of Section 6.1 we made two modeling choices pertaining to (i) representations of the papers provided to similarity-computation algorithms, and (ii) the size of reviewers' profiles used by these algorithms. In this section, we investigate these two questions in more detail.

- *Question 1 (paper representation).* Some similarity-computation algorithms are designed to work with titles and/or abstracts of papers (e.g., Specter) while others can also incorporate the full texts of the manuscripts (e.g., TPMS). Consequently, there is a potential trade-off between accuracy and computation time. Richer representations are envisaged to result in higher accuracy, but are also associated with an increased demand for computational power. As with the choice of the algorithm itself, there is no guidance on what amount of information should be supplied to the algorithm as the gains from using more information are not quantified. With this motivation, our first question is:

  *What are the benefits of providing richer representations of papers to similarity-computation algorithms?*

- *Question 2 (reviewer profile).* The second important choice is the size of reviewers' profiles. On the one hand, by including only very recent papers in reviewers' profiles, conference organizers are at risk of not using enough data to obtain accurate values of expertise. On the other hand, old papers may not accurately represent the current expertise of researchers and hence may result in noise when used to compute expertise. Thus, our second question is:

  *What is the optimal number of the most recent papers to include in the profiles of reviewers?*

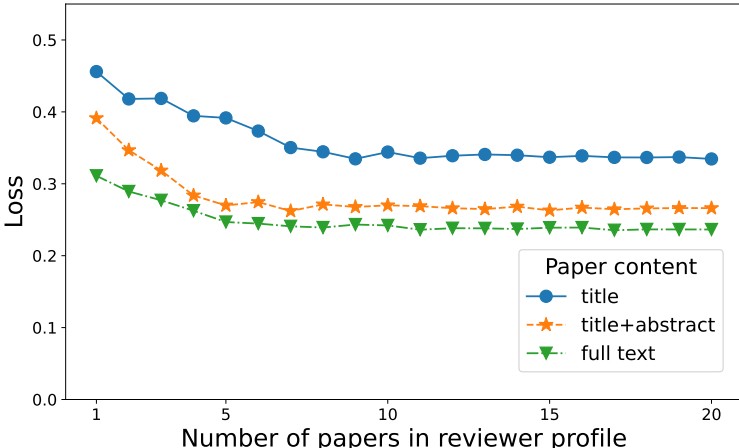

Figure 2: Impact of different choices of parameters on the quality of predicted similarities. Confidence intervals are not shown (see Table 4).

| Paper Representation | Loss | 95% CI for Loss | Δ with Title+Abstract | 95% CI for Δ |
|---|---|---|---|---|
| Title | 0.33 | [0.29, 0.38] | +0.07 | [0.02, 0.12] |
| Title+Abstract | 0.27 | [0.22, 0.32] | — | — |
| Full Text | **0.24** | [0.19, 0.30] | −0.03 | [−0.09, 0.03] |

Table 4: Performance of the TPMS algorithm with 20 most recent papers included in reviewers' profiles and with different choices of the paper representation. Lower values of loss are better. A positive (respectively, negative) value of Δ indicates that the corresponding choice of the paper representation leads to a weaker (respectively, stronger) performance.

To investigate these questions, we choose the TPMS algorithm as the workhorse to perform experiments. We make this choice for two reasons. First, TPMS can work with all possible choices of the paper representation: title only, title and abstract, and full text of the paper. In contrast, other methods do not support using the full text of the papers. Second, TPMS is fast to execute, enabling us to compute expertise for hundreds of configurations in a reasonable time.

With the algorithm chosen, we vary the number of papers in the reviewers' profiles from 1 to 20. For each value of the profile length, we consider three representations of the paper content: title, title+abstract, and full text of the paper. For each combination of parameters, we construct reviewer profiles and predict similarities using the approach introduced in Section 6.1. The only exception is that we repeat the procedure for averaging out the randomness in the profile creation 5 times (instead of 10) to save the computation time.

**Results** Figure 2 and Table 4 provide answers to the questions we study in this section. Figure 2 shows the pointwise loss of the TPMS algorithm for each choice of parameters. To save computation time, we do not build confidence intervals for each combination of parameters. Instead, Table 4 sets the number of papers in reviewers' profiles to 20 (consistent with Table 3) and presents confidence intervals for losses incurred by the algorithm under different choices of paper representations. We now make two observations.

First, paper abstracts are very useful in improving the quality of expertise prediction as compared to titles alone (the improvement of 0.07). Including full texts of the papers in reviewers' and papers' profiles results in additional improvement (0.03). Overall, ignoring the minor differences in the datasets between this section and Section 6.1, we observe that TPMS with titles, abstracts, and full texts of papers demonstrates the same performance as SPECTER+MFR which cannot handle the full texts of papers. Thus, it may be of interest to develop modern deep-learning based algorithms to incorporate full texts of papers and further boost performance on the similarity-computation task.

Second, the loss curves plateau once reviewer profiles include 8 or more of their most recent papers. Additional increase of the profile length does not impact the quality of predictions. Thus, in practice, reviewers may be instructed to include 10 representative papers to their profile, which for many active researchers amounts to the number of papers published in 1-3 years.

## 7 Additional evaluations

Recall that our loss metric (see Section 5) assigns different weights to errors of an algorithm: errors on pairs with similar values of reviewer's expertise are penalized less than errors on well-separated pairs. The overall performance of the algorithm is then captured in a single number (loss) which does not characterize the type of mistakes made by the algorithm.

To provide deeper characterization of the algorithm performance, we now conduct additional evaluations. In these evaluations, we focus on two important regions of the similarity-computation problem. Specifically, from all expertise evaluations made by participants, we select two groups of triples, where each triple consists of a participant $r$ and two papers $(p_1, p_2)$ evaluated by this participant:

- **"Easy" triples.** The first group comprises 261 triples where a participant reported high expertise (greater than or equal to four) in one paper and low expertise (less than or equal to two) in another.

  We call these triples "easy" as the gap in the expertise between the two papers is large. Note that in real conferences it is important to resolve the easy triples correctly to ensure that reviewers do not get assigned irrelevant papers.

- **"Hard" triples.** The second group comprises 417 triples where a participant reported high expertise (greater than or equal to four) in both papers and the values of expertise are different for the two papers.

  We call these triples "hard" as the gap in the expertise between the papers is small. That said, we note that resolving hard triples in practice is also very important: in real conferences, we want to assign papers to the most suitable reviewers and we need to be able to distinguish two papers with high but not equal values of expertise to achieve this goal.

Note that by focusing on these two groups, we exclude regions of the problem which may be considered less important. For example, the ability of an algorithm to correctly order two low-expertise papers is less crucial as long as the algorithm can reliably distinguish these papers from the high-expertise papers.

We now study the performance of the algorithms introduced in Section 5.2 on the easy and hard groups of triples. For all algorithms we use titles and abstracts of papers as the data source and include 20 papers in the reviewers' profiles. Specifically, for each triple we compare the ordering of papers predicted by an algorithm with the expertise-induced ordering. We then compute the accuracy of each algorithm as the fraction of triples correctly resolved by the algorithm (from 0 to 1, larger values are better). In addition, we also evaluate TPMS in the full text regime to estimate the effect of papers' texts on the accuracy of the similarity-computation algorithms.[4]

Table 5 demonstrates the results of additional evaluations. First, note that all algorithms have moderate to high accuracy in resolving the easy triples. All of the algorithms except ELMO detect papers that reviewers have low expertise in with probability close to or above 80%, with SPECTER+MFR reaching nearly 90% accuracy. However, we note that in a non-trivial amount of cases (more than 10%), the algorithms are unable to distinguish a paper that a reviewer is well-qualified to review versus a paper the reviewer does not have necessary background to evaluate. Thus, there is a vital need to improve the similarity-computation algorithms.

Getting to the hard triples, we observe a significant drop in performance with all algorithms being a bit better than random: the best-performing algorithm (TPMS) reaches 62% accuracy in the title+abstract regime and 64% in the full text regime. Perhaps surprisingly, while SPECTER+MFR and SPECTER outperform TPMS

---

[4]TPMS in the full text regime is evaluated on a slightly different dataset as explained in Section 6.2.

|  | Easy triples ($n = 261$) | | Hard triples ($n = 417$) | |
| --- | --- | --- | --- | --- |
| Algorithm | Accuracy | 95% CI | Accuracy | 95% CI |
| TPMS (T+A) | 0.80 | [0.72, 0.87] | 0.62 | [0.54, 0.69] |
| TPMS (Full Text) | 0.84 | [0.76, 0.91] | **0.64** | [0.56, 0.70] |
| ELMo | 0.70 | [0.62, 0.78] | 0.57 | [0.51, 0.63] |
| Specter | 0.85 | [0.76, 0.92] | 0.57 | [0.50, 0.63] |
| Specter+MFR | **0.88** | [0.81, 0.94] | 0.60 | [0.53; 0.66] |
| ACL | 0.78 | [0.69; 0.86] | 0.62 | [0.55; 0.68] |

Table 5: Results of additional evaluations. Higher values of accuracy are better. Specter+MFR demonstrates the best performance on easy triples and TPMS (Full Text) has the highest accuracy on hard triples.

on easy triples, they are not better than TPMS on hard triples (we caveat though that error bars are too wide to make a decisive comparison). This observation suggests that there may be a value in additionally fine-tuning these advanced algorithms on hard triples to improve their practical performance.

Finally, we observe that on both easy and hard triples, full texts of the papers are instrumental in improving performance of the TPMS algorithm. This observation supports our intuition that similarity-computation algorithms do indeed benefit from employing full texts of papers.

With these remarks, we conclude evaluations of algorithms on the dataset we collected.

## 8 Discussion

In this work, we collect a novel dataset of reviewers' expertise in reviewing papers. In contrast to datasets collected in previous works, our dataset (i) is released publicly, and (ii) contains evaluations of expertise made by scientists who have actually read the papers for their own research purposes. We use this dataset to compare several existing expertise-estimation algorithms and help venue organizers in choosing an algorithm in an evidence-based manner.

Most importantly, we find that current algorithms make a large number of errors. Given the growing number of submissions in many fields of research, the need for automation in reviewer assignments is only growing. There is thus an urgent and vital need to develop significantly improved algorithms to match reviewers to papers, thereby in turn making the peer-review process considerably better.

Our dataset can be used to develop as well as evaluate new expertise-estimation algorithms. We encourage researchers from the natural language processing and other communities to use our data in order to improve peer review.

**Limitations.** Finally, we mention several caveats that researchers should be aware of when working with our dataset and interpreting the results of our experiments. First, our dataset comprises evaluations of expertise in reviewing papers that were written some time ago. In contrast, in real conferences, many papers are recent and not available online. Thus, incoming citations to papers included in our dataset may constitute information that is not available to the algorithms in real life. While the algorithms we evaluate in this work do not rely on the citation relationship, this caveat may be important for future work.

Second, the experiments we conduct in this work rely on Semantic Scholar profiles. These profiles are constructed automatically and may not be accurate. Thus, mistakes of the algorithms we observe in this work can be partially due to the noise in the profile creation.

Finally, we reiterate that the present version of the dataset was constructed by participants who collectively are not representative of the general computer science community. For example, about 40% of participants are affiliated with a certain university. To alleviate this issue, we continue the data collection process and encourage the readers of this paper to contribute their data points to the dataset:

*(link redacted for double blind reviewing; survey replicated in Appendix A)*

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

# Appendices

We now provide additional discussion.

## A   More details on the survey used for data collection

In this section we provide full instructions that were given to the participants of our data-collection procedure.

```
                    Dataset of Reviewing Expertise

The goal of this experiment is to collect a dataset to help researchers design better
algorithms for computing similarities between papers and reviewers:  these algorithms
will help to improve matching of reviewers to papers in many conferences like
NeurIPS, AAAI, ACL, etc.

WHAT DO YOU NEED TO DO?
***Recall 5-10 papers in your broad research area that you read to a reasonable
extent in the last year and tell us your expertise in reviewing these papers.***
(HINT: To quickly recall what papers you read recently, you can search for arxiv.org
or an analogous website in your browser history.)

WHICH PAPERS TO REPORT?

  • Papers should not be authored by you

  • Papers should be freely available online (preferably arXiv, but other open
    sources are also fine)

**Suggestions**

  • Try to choose a set of papers such that some pairs are well-separated and some
    are very close in terms of your expertise

  • Please try to avoid ties in the expertise ratings you provide

  • Try to think of some papers that are less famous to make the dataset more diverse

  • Try to provide some examples that are not obvious and may be tricky for the
    similarity-computation algorithms.  For example, a naive computation of
    similarity may think that a paper on "Theory of Information Dissemination in
    Social Networks" has high similarity with an Information Theory researcher, but
    in reality, this researcher may not have expertise in reviewing this paper
```

```
WHAT PARTS OF DATA YOU PROVIDE WILL BE RELEASED?
To facilitate the development of better algorithms for similarity computation
(trained to perform well on your data!), we will publicly release data collected in
this survey (except email addresses) in a non-anonymized form.  Your email will not
be released.

WHO IS RUNNING THIS SURVEY?
(Redacted for double blind review.)

[...]

List up to 10 papers in your broad research area that you read to a reasonable extent
in the last year and tell us your expertise in reviewing these papers.  Please try to
enter at least 5 papers.

Link to Paper 1:________

Expertise in reviewing Paper 1:

  • 1.0 (I am not qualified to review this paper)
  • 1.25
  • 1.5
  • 1.75
  • 2.0 (I can review some aspects of the paper, but can't make a reliable overall
    judgment)
  • ...
  • 3.0 (I can provide an adequate review, but a substantial part of the paper is
    outside of my expertise)
  • ...
  • 4.0 (I have background in most aspects of the paper, but some minor aspects are
    beyond my expertise)
  • ...
  • 5.0 (I have background necessary to evaluate all the aspects of the paper)

[...]
```

## B   More details on the ACL algorithm

In this section, we provide more details on the ACL algorithm that we evaluate in this paper.

**Training**   The model is trained on a large corpus of 45,309 abstracts from the ACL anthology and is inspired by the work of Wieting et al. (2019; 2022). Specifically, the model optimizes a max-margin contrastive learning objective which is defined as follows. First, each abstract $a_i$ from the corpus is split into two disjoint segments of text uniformly at random. These segments are then uniformly at random allocated into two equally-sized groups: $a_i^{(1)} \in A_1$ and $a_i^{(2)} \in A_2$.

Second, positive and negative examples are constructed as follows:

- *Positive example:* For each abstract $a_i$, pair $(a_i^{(1)}, a_i^{(2)})$ constitutes a positive example.
- *Negative example:* For each passage $a_i^{(1)} \in A_1$, a counterpart $t_i \neq a_i^{(2)}$ from $A_2$ is selected to maximize the notion of cosine similarity

$$f_\theta(a_i^{(1)}, t_i) = \cos\left(g(a_i^{(1)}, \theta), g(t_i, \theta)\right),$$

where $g$ is the sentence encoder with parameters $\theta$. Pair $(a_i^{(1)}, t_i)$ constitutes a negative example.

Finally, with this procedure to build positive and negative examples, the objective of the ACL algorithm is defined as:

$$\min_{\theta} \sum_i \left[ \delta - f_\theta(a_i^{(1)}, a_i^{(2)}) + f_\theta(a_i^{(1)}, t_i)) \right]_+ {}^5$$

Inner-working of the algorithm relies on `sentencepiece` embeddings[6] (Kudo & Richardson, 2018) with dimension of 1,024 and vocabulary size of 20,000. The encoder, $g$, simply mean pools the learned `sentencepiece` embeddings, making for efficient encoding, even on CPU.[7] In the training procedure, a batch size of 64 is used and the model is trained for 20 epochs. The margin, $\delta$, is set to 0.4.

**Inference** At the inference stage, for a given pair of a submission and a reviewer, the similarity score is defined as a combination of cosine similarities between the submission's abstract and three most-similar abstracts from the reviewer's profile. Specifically, let $s_1, s_2$ and $s_3$ be the top-3 cosine similarities between the submission's abstract and abstracts from the reviewer's profile. The similarity score between the submission and the reviewer is then defined as follows:

$$s = s_1 + \frac{s_2}{2} + \frac{s_3}{3}.$$

If a reviewer has less than three abstracts in the profile, the corresponding cosine similarity scores $s_i$ are set to be zero.

---

[5]We use $[\cdot]_+$ to denote function $h : h(x) = \max(x, 0)$.
[6]https://github.com/google/sentencepiece
[7]See Wieting et al. (2019; 2022) for more details on encoding speed.

