# OpenReview forum: "A Gold Standard Dataset for the Reviewer Assignment Problem"
_TMLR — Rejected by TMLR_

### Review · Reviewer_uEkS · 2024-02-08

**Summary Of Contributions:**

The paper introduces a new dataset that includes expertise scores from researchers to help improve how academic papers are matched with reviewers. This dataset is significant as it allows for a more nuanced approach to determining reviewer suitability beyond simple keyword matching. The paper tests several algorithms to see how well they can predict good matches based on this dataset, employing a variety of methods to assess the accuracy and relevance of the matches. The paper finds that one algorithm, Specter+MFR, works best for using titles and abstracts to make these predictions, highlighting the potential of machine learning techniques in enhancing the peer review process. This work helps in developing better tools for assigning reviewers to papers and provides insights regarding improving existing algorithms for peer review assignment, suggesting future directions for research in this area.

**Audience:**

Yes

**Broader Impact Concerns:**

No concerns.

**Claims And Evidence:**

Yes

**Requested Changes:**

1. **Address Potential Biases**: The authors should explore methodologies to reduce the impact of biases in self-reported expertise scores. For instance, cross-referencing these self-assessments with objective indicators of expertise, such as the number and impact of publications and citations, could provide a more balanced view. Additionally, incorporating peer assessments, where researchers' expertise is evaluated by their colleagues, could offer a complementary perspective that mitigates individual biases. This multifaceted approach would enhance the reliability of the dataset and, by extension, the validity of the algorithm evaluations.

2. **Incorporate Additional Evaluation Metrics**: Expanding the set of evaluation metrics can provide a deeper understanding of each algorithm's performance across various dimensions. Beyond matching accuracy, it's important to consider how well these algorithms manage reviewer workload—ensuring no reviewer is overwhelmed or underutilized. Assessing the diversity of assigned reviewers in terms of expertise and demographics is crucial for promoting inclusive and comprehensive reviews. Moreover, evaluating the timeliness of the review process—how quickly and efficiently reviewers are matched to papers—can highlight algorithms that optimize for speed without sacrificing quality.

3. **Discussion on Algorithm Scalability and Privacy Concerns**: As academic conferences grow in size and scope, the scalability of reviewer assignment algorithms becomes increasingly critical. The paper should address how these algorithms perform under the stress of large datasets and whether their computational complexity is manageable at scale. Privacy concerns are paramount, given the sensitive nature of unpublished research. The authors should discuss measures to protect submission data, such as encryption and anonymization techniques, and how these considerations are integrated into the algorithmic design to safeguard researchers' work.

**Strengths And Weaknesses:**

The paper is well written and puts itself nicely in context of past work.

**Strengths:**

1. **Valuable Dataset Creation**: The introduction of a gold standard dataset specifically designed for the reviewer assignment problem is a notable advancement for the academic community. By focusing on self-reported expertise scores, the dataset provides a nuanced perspective that goes beyond traditional keyword-based matching. This allows for a more accurate and personalized approach to determining the fit between reviewers and papers, offering a critical tool for both testing and improving algorithmic solutions for academic peer review. The inclusion of self-reported data introduces a human element into the algorithmic matching process, providing depth and complexity that purely computational methods might miss.

2. **Comprehensive Algorithm Evaluation**: The paper's extensive evaluation of multiple algorithms against this dataset sets a benchmark in the field, offering a detailed comparison of their effectiveness in matching reviewers with papers. This thorough analysis is essential for identifying the strengths and limitations of existing approaches, serving as a valuable resource for researchers and developers aiming to enhance the peer-review process. By elucidating the performance of different algorithms, the paper facilitates a deeper understanding of how various computational techniques can be optimized to improve the accuracy and efficiency of reviewer assignments, thereby streamlining the academic publication process.

3. **Insightful Findings**: The discovery that the Specter+MFR algorithm excels in making predictions based on titles and abstracts signals a promising direction for future research, emphasizing the utility of deep learning and natural language processing in academic contexts. This insight underscores the potential for more sophisticated text analysis methods to significantly improve the matching process, suggesting that algorithms capable of understanding the semantic content of papers could outperform more traditional, keyword-based approaches. Additionally, the observation that a relatively straightforward method like TF-IDF, as utilized by TPMS, remains highly competitive with complex embedding-based strategies, highlights the importance of baseline approaches and the nuanced trade-offs between simplicity and sophistication in algorithm design. These findings enrich our understanding of algorithmic capabilities and limitations, guiding future efforts to refine and develop more effective solutions for the peer-review system.

**Weaknesses:**

1. **Scope Limitation to Computer Science**: The dataset's exclusive focus on computer science limits its direct applicability to other fields, which might have different criteria for evaluating expertise and matching reviewers to papers. This specialization restricts the dataset's utility for interdisciplinary research areas where the boundaries of expertise are more fluid and the content more varied. Expanding the dataset to include more diverse academic disciplines could enhance its versatility and allow for a more comprehensive understanding of how different fields approach the peer-review process.

2. **Potential Biases in Self-Reported Data**: The reliance on self-reported expertise introduces the risk of biases, such as overestimation or underestimation of one's own abilities, influenced by factors like confidence levels, memory recall, and social desirability. These biases could skew the dataset, leading to inaccuracies in matching algorithms. Without mechanisms to verify or normalize these self-assessments, the dataset may not fully represent the range of expertise or the nuanced differences in reviewer capabilities across the research community.

3. **Lack of Consideration for Broader Reviewer Assignment Challenges**: Focusing solely on matching accuracy based on expertise overlooks other critical dimensions of the reviewer assignment process. For example, the distribution of review workloads is a key concern, as imbalances can lead to delays and affect the quality of reviews. Additionally, the diversity of reviewers, in terms of geographic location, gender, academic background, and seniority, plays a crucial role in ensuring a broad and inclusive perspective on the reviewed work. Addressing these factors is essential for a fair and effective peer-review system.

4. **Absence of a Novel Solution Algorithm**: The paper's analysis provides valuable insights into the performance of existing algorithms but stops short of offering a new algorithmic solution. This omission represents a missed opportunity to advance the field by tackling identified shortcomings directly. By not proposing an innovative solution that addresses the gaps and challenges highlighted through the dataset and algorithm comparisons, the paper leaves the development of more effective and efficient matching algorithms to future work. The current contributions of the paper are a little thin and it would have been great if it proposed an algorithmic solution also.

---

> ### Author Response · Authors · 2024-03-12
> **Response to reviewer uEkS**
>
> We thank the reviewer for their careful reading of the paper and their detailed enumeration of the strengths and weaknesses of the paper. Thank you very much for your thoughtful comments. We agree that a limitation of this work is the diversity of the respondents of the dataset, in particular geographical diversity. We did try our best to recruit diverse participants and the survey was open to all, however the current set of participants is what we eventually obtained. While in the initial submission this was highlighted in Section 4, in the revision, we have also highlighted this limitation in the introduction: "as we discuss later in more detail, the dataset is not devoid of biases, with bias in terms of geographical skew of participants being most prominent." As we also discuss in the introduction, we do call on  people to participate in this survey even after publication, and help make the population more diverse. We will give the link to the participation form in the camera ready version, which is currently redacted for dual anonymous reviewing, and we envisage updating the dataset periodically accordingly. We also thank the reviewer for the additional suggestions. These are indeed very interesting, but are beyond the scope of this paper.

---

### Review · Reviewer_jZZ1 · 2024-02-19

**Summary Of Contributions:**

This paper proposes a dataset for evaluating reviewer assignment algorithms. This consists of self-reported expertise scores from a pool of computer science researchers. The authors also use their dataset to evaluate baseline paper matching algorithms with different features. Their key finding is that TPMS performs surprisingly well when used on the full paper text. It is competitive with complex neural representations such as SPECTER + MFR which only use the paper title, abstract, and citation graph. The paper also identifies a large drop in performance of all algorithms when evaluated on "hard" (participant, paper_1, paper_2) similarity triples compared to "easy" examples.

**Audience:**

Yes

**Broader Impact Concerns:**

This paper sufficiently addresses concerns about the potential bias in their data collection process (40% of participants from 1 university, only 25% of papers published before 2020).

**Claims And Evidence:**

Yes

**Requested Changes:**

### Critical
- Use the filtered dataset in Section 6.2 throughout the paper (Sections 6.1 and 7).
- Compare $\Delta$ with TPMS (Title + Abstract) in Table 5, similar to Tables 3-4.
- Show table results with 1 standard deviation.
- Mention more details about survey instructions in the main text (include some "less famous" and "tricky" examples).

### Non-Critical
- Use larger pre-trained models and/or fine-tune on data from scientific domains, as suggested in Section 6.1. Running these experiments would provide more support for the recommendation to develop deep learning models that include full paper text in the input representation.
- Comment on the effect of calibration/normalization of scores across participants.

**Strengths And Weaknesses:**

### Strengths
- Novelty: Cites and compares to related work where appropriate.
- Experimental Setup: Good analysis/ablation over relevant aspects of the problem (number of reviewer papers, paper representations, easy vs. hard triples).
- Impact: Substantial contributions in a well-motivated problem area.
- Clarity: well-organized and well-written.

### Weaknesses
- Consistency: Depending on the section, the experimental results use 2 slightly different versions of the dataset. The paper's claims would be much stronger if comparisons were made using the exact same dataset for all experiments. This can be fixed by using the dataset from Section 6.2 everywhere, i.e. removing the handful of papers which do not have pdf text freely available or are not listed on Semantic Scholar. Additionally, "low expertise" is defined as $\leq 2$ in Section 4.3, and as $\leq 3$ in Section 7.
- Results: Many of the 95% confidence intervals are overlapping, which makes it difficult to draw meaningful conclusions. This is emphasized more clearly in some sections than others. Since the 95% CI represents approximately 2 standard deviations, it would be good to see if using 1 standard deviation provides any additional (albeit lower confidence) conclusions.

### Typos
- Table 1 caption: "first four characteristics" should be "first three characteristics".
- Section 5.2: "variety downstream tasks" should be "variety of downstream tasks".

---

> ### Author Response · Authors · 2024-03-12
> **Response to reviewer jZZ1**
>
> Thank you very much for your careful and detailed reading of the paper and very helpful suggestions.
>
> Regarding your point on consistency:
>
> Thank you for pointing this out. We adjusted the experiments (and corresponding explanation) so that the datasets are consistent.
> Because the number of papers and reviewers changed minimally, the results stayed almost the same. The only "loss" value that resulted in a change in the paper was ELMo changing from 0.34 to 0.35, and no quantity in the table was modified by more than 0.02. The main text in the paper stayed the same.
>
> Regarding your comment on "low expertise" difference:
>
> We apologize for the confusion. In the revision, we have provided a complete counts of each response value (Figure 1a), and stated that about 56\% of reported expertise scores are 4 or higher.
>
>
> Regarding your suggestion about the 1-standard deviation error bars:
>
> We recomputed the tables with the 1-standard deviation. Here are the results. We are not sure about including them in the paper as 1-standard deviation is not common, so we thought it is best to leave it as a publicly accessible comment here but not in the manuscript. We are open to any alternative suggestions from the reviewer on this.
>
> Table 3 (now specter_mfr's pairwise confidence intervals are not overlapping with tpms)
>
>         tpms: 0.28 	  [0.25; 0.3]
>        elmo: 0.35 	 [0.33; 0.39] 	  0.07 	  [0.05; 0.11]
>     specter: 0.27 	 [0.24; 0.31] 	 -0.01 	 [-0.03; 0.03]
> specter_mfr: 0.24 	 [0.21; 0.27] 	 -0.04 	 [-0.07; -0.01]
>         acl:  0.3 	 [0.27; 0.33] 	  0.02 	   [0.0; 0.05]
>
> Table 4
>
>   t: 0.33 	 [0.31; 0.36] 	 0.07 	  [0.04; 0.09]
> ta: 0.27 	  [0.24; 0.3]
>  f: 0.24 	  [0.2; 0.27] 	 -0.03 	 [-0.06; -0.0]
>
> Table 5 Easy Triples
>
>         tpms:  0.8 	 [0.76; 0.84]
>        elmo:  0.7 	 [0.66; 0.74]
>     specter: 0.85 	 [0.81; 0.89]
> specter_mfr: 0.88 	 [0.84; 0.91]
>         acl: 0.78 	 [0.74; 0.82]
>      tpms_f: 0.84 	  [0.8; 0.88]
>
> Table 5 Hard Triples
>
>         tpms: 0.62 	 [0.58; 0.66]
>        elmo: 0.57 	  [0.54; 0.6]
>     specter: 0.57 	  [0.53; 0.6]
> specter_mfr:  0.6 	 [0.56; 0.64]
>         acl: 0.62 	 [0.58; 0.65]
>      tpms_f: 0.64 	  [0.6; 0.68]
>
> Regarding "Mention more details about survey instructions in the main text (include some "less famous" and "tricky" examples).":
>
> Based on your suggestion, we have now included the text provided to the participants containing an example of what we meant by tricky examples. This is in the "expertise evaluations" part of Section 3 in the revision.

---

### Review · Reviewer_esF1 · 2024-02-21

**Summary Of Contributions:**

- This work contributes a dataset with real reviewer annotations for the task of reviewer assignment in top conferences.
- It benchmarks the state-of-the-art reviewer matching algorithms that use abstract and title and shows that the best algorithm has similar performance to the simple TPMS with full text.
- It conducts analysis to show what potential improvement can be added to existing algorithms.

**Audience:**

Yes

**Claims And Evidence:**

Yes

**Requested Changes:**

- It would be great to perform this study on a more diverse population, including more non-US reviewers.
- It would be better to have some ablation studies to show the effectiveness of the design choices. For example, with a different survey, the quality of self evaluation drops.

**Strengths And Weaknesses:**

Strength
- the dataset contains high-quality self evaluation of expertise with a carefully survey.
- comprehensive benchmark of existing methods.

Weakness
- I am not sure how relevant is this paper to TMLR.
-  The total numbers of participants and papers are quite small and most of them are from the US, which is not really representative.
-  No discussion on the possible selection bias of the papers recalled by the reviewers, given the fact that some reviewers may have reviewed much more than 5-10 papers. Also there is no discussion on the difference between different roles as reviewers, such as PC, SPC or Meta Reviewer etc.
- It seems to me the 0.25 step size in evluation score is too fine-grained for people to give an accurate score. Maybe an ordered list is good enough.

---

> ### Author Response · Authors · 2024-03-12
> **Response to reviewer esF1**
>
> Thank you very much for your thoughtful comments. We agree that a limitation of this work is the diversity of the respondents of the dataset, in particular geographical diversity. We did try our best to recruit diverse participants and the survey was open to all, however the current set of participants is what we eventually obtained. While in the initial submission this was highlighted in Section 4, in the revision, we have also highlighted this limitation in the introduction: "as we discuss later in more detail, the dataset is not devoid of biases, with bias in terms of geographical skew of participants being most prominent." As we also discuss in the introduction, we do call on  people to participate in this survey even after publication, and help make the population more diverse. We will give the link to the participation form in the camera ready version, which is currently redacted for dual anonymous reviewing, and we envisage updating the dataset periodically accordingly.
>
> Regarding your point on "It seems to me the 0.25 step size in evaluation score is too fine-grained for people to give an accurate score. Maybe an ordered list is good enough."  In our experiments, we have used rankings as a measure of accuracy. Specifically, in Section 7, conditioned on whether we are in the easy or hard triple setting, the loss function is the 0-1 loss on whether the algorithm predicts the relative ranking of a pair of papers correct or not.
>
> Finally, we think that this paper is of interest to a significant fraction of the machine learning community (and hence to TMLR) for at least two reasons:
>
> 1. The machine learning and natural language processing communities are at the forefront of developing algorithms for assigning reviewers. Such a dataset is thus directly helpful for them in this development and/or evaluation.
>
> 2. The entire review process at machine learning venues like NeurIPS/ICML/ICLR as well as at TMLR is highly affected by the accuracy of algorithms that compute reviewer expertise. Thus naturally it is of interest to the community to understand the benefits and blindspots of these algorithms.

---

### Decision · Action_Editor_Ye1J · 2024-04-13

**Recommendation:** Reject

**Comment:**

Although the authors had requested additional time for revisions, in discussion with TMLR editors-in-chief it was felt a decision could be made at this point.

Even after considering author responses, the reviewers are divided in their opinion of this paper. Two reviewers are leaning towards rejection and one towards acceptance.

Several positive aspects were noted:
+ Contributing a high-quality dataset with real reviewer annotations was appreciated [esF1,uEkS]
+ The comparison to related work was appreciated [jZZ1]
+ The comprehensive benchmark of methods was appreciated [esF1,uEkS]
+ Some of the findings were considered insightful [uEkS]
+ Ablation over numbers of reviewer papers, representations, and easy/hard triples was appreciated [jZZ1]
+ The manuscript was considered well organized and written [jZZ1]

However, several concerns were pointed out by the reviewers along different aspects.

In overall comments:
- Relevance to TMLR was questioned [esF1]. Authors provided some arguments in favor of the relevance.
- Lack of a new algorithmic solution was criticized [uEkS]

A number of concerns regarding the scope of the study were pointed out:
- Limitation to the computer science field was criticized [uEkS]
- The small number of participants was criticized [esF1]
- Having participants mostly only from the US was criticized [esF1]
- Lack of discussion of possible selection bias was criticized [esF1]. Authors pointed out some places where they have mentioned it.

Several concerns were noted in how the analysis was carried out:
- Potential bias in self-reported reviewer expertise was a concern [uEkS]
- Lack of discussion of different reviewer roles was criticized [esF1]
- Lack of consistency due to using two different data set versions and two different definitions of low expertise was criticized [jZZ1]. Authors commented they ran an adjusted experiment with little change in results.
- Lack of clear conclusions due to overlapping confidence intervals was criticized [jZZ1]. Authors provided some tables with 1-standard deviation intervals.
- The evaluation score step size was criticized as too fine grained [esF1]. Authors responded they used a ranking criterion too in evaluation, however the reviewer may have meant it was hard for reviewers yielding noise.
- Discussion of calibration/normalization was desired [jZZ1]

A number of suggestions for expanding the work were made:
- Additional ablation studies for the design choices were desired [esF1]
- Use of larger pretrained models finetuned on scientific domains was desired [jZZ1]
- Limitation to the matching accuracy was criticized and additional metrics considering also aspects of diversity and reviewer workload were desired [uEkS]. Authors seemed to consider this beyond the scope of the paper.
- Discussion of algorithm scalability and privacy was desired [uEkS]. Authors seemed to consider this beyond the scope of the paper.

Based on the reviews at least some people in TMLR's audience would certainly be interested in the work. However, whether the claims are supported by accurate, convincing and clear evidence is in question due to the concerns of the reviewers. The concerns regarding the scope of the study, and the concerns about the analyses including handling of biases, are especially important if the manuscript aims to provide a "gold standard dataset" for reviewer assignment.

It thus seems clear that although there is interest in the work, the work could not be currently accepted either as is or with minor revision. Thus I must recommend rejection at this stage. It could be that after the authors have carried out further revisions the work could become acceptable and could be resubmitted.

**Audience:**

It seems clear from the reviews that at least some people in TMLR's audience would be interested in the work.

**Claims And Evidence:**

Whether the claims are supported by accurate, convincing and clear evidence is in question due to the concerns of the reviewers. The concerns regarding the scope of the study, and the concerns about the analyses including handling of biases, are especially important if the manuscript aims to provide a "gold standard dataset" for reviewer assignment. See the Comment field for more detailed discussion.

**Resubmission Of Major Revision:**

The authors may consider submitting a major revision at a later time.